Journal of Data-centric Machine Learning Research (2026)          Submitted 6/25; Revised 10/25; Published 3/26

# The CLRS-Text Algorithmic Reasoning Language Benchmark

**Larisa Markeeva**[*]                                                                LMARKEEVA@GOOGLE.COM
*Google DeepMind*
*14-18 Handyside Street, London, N1C4DN, UK*

**Sean McLeish**[*]                                                                      SMCLEISH@UMD.EDU
*University of Maryland*
*College Park, MD 20742, USA*

**Borja Ibarz**[*]                                                                          BIBARZ@GOOGLE.COM
*Google DeepMind*
*14-18 Handyside Street, London, N1C4DN, UK*

**Wilfried Bounsi**                                                                      WILCOLN@GOOGLE.COM
*Google DeepMind*
*14-18 Handyside Street, London, N1C4DN, UK*

**Olga Kozlova**                                                                          GRENLAYK@GOOGLE.COM
*Google DeepMind*
*14-18 Handyside Street, London, N1C4DN, UK*

**Alex Vitvitskyi**                                                                        AVLIFE@GOOGLE.COM
*Google DeepMind*
*14-18 Handyside Street, London, N1C4DN, UK*

**Charles Blundell**                                                                    CBLUNDELL@GOOGLE.COM
*Google DeepMind*
*14-18 Handyside Street, London, N1C4DN, UK*

**Tom Goldstein**[†]                                                                      TOMG@UMD.EDU
*University of Maryland*
*College Park, MD 20742, USA*

**Avi Schwarzschild**[†]                                                          SCHWARZSCHILD@CMU.EDU
*Carnegie Mellon University*
*5000 Forbes Avenue Pittsburgh, PA 15213, USA*

**Petar Veličković**[†]                                                                  PETARV@GOOGLE.COM
*Google DeepMind*
*14-18 Handyside Street, London, N1C4DN, UK*

**Reviewed on OpenReview:** *https://openreview.net/forum?id=1sZ8RsVEoX*

**Editor:** Christopher De Sa

## Abstract

Eliciting reasoning capabilities from language models (LMs) is a critical direction on the path towards building intelligent systems. Many foundational studies dedicated to reason-

ing focus on out-of-distribution performance on procedurally-generated synthetic benchmarks, bespoke-built to evaluate specific skills only. This trend makes results hard to transfer across publications, slowing down progress. Several years ago, a similar issue was identified and rectified in the field of neural algorithmic reasoning, with the advent of the *CLRS* benchmark. CLRS is a *dataset generator* comprising graph execution traces of classical algorithms from the Introduction to Algorithms textbook. Inspired by this, we propose **CLRS-Text**—a textual version of these algorithmic traces. Out of the box, CLRS-Text is capable of procedurally generating trace data for thirty diverse, challenging algorithmic tasks across any desirable input distribution, while offering a standard pipeline in which any additional algorithmic tasks may be created in the benchmark. We fine-tune and evaluate various LMs as generalist executors on this benchmark, validating prior work and revealing a novel, interesting challenge for the LM reasoning community. Our code is available at `https://github.com/google-deepmind/clrs/tree/master/clrs/_src/clrs_text`.

**Keywords:** large language models, algorithmic reasoning, out-of-distribution generalisation, length generalisation, multi-task learning, random positional embeddings.

## 1 Introduction

In spite of the impressive performance of language models (LMs) in a variety of scenarios (Team et al., 2024; OpenAI, 2023), they also exhibit some well-documented failures, especially when it comes to *robust reasoning* (Dziri et al., 2024). Some examples of this relate to reverse concepts (Berglund et al., 2023), elementary arithmetic (Shen et al., 2023; Zhou et al., 2024) and geometry (Mouselinos et al., 2024), or recognising higher-order languages (Delétang et al., 2022). Improving base LM performance on reasoning tasks is important, as their present unreliability makes them unwieldy to use in open-ended environments requiring robust behaviours, e.g. multi-step planning and scientific problems.

To evaluate the robustness of reasoning capabilities of LMs, we frequently employ static datasets requiring mathematical reasoning (Cobbe et al., 2021; Hendrycks et al., 2021) or QA (Liu et al., 2020; Rein et al., 2023), but it is well-understood that hill-climbing on any *static* dataset may result in an illusion of progress, especially given vast quantities of data available in pre-training corpora. Even resampling of values on these benchmarks can result in drastic regressions (Mirzadeh et al., 2024)—a phenomenon known as the *reasoning gap* (Srivastava et al., 2024). While it is not yet completely certain to what extent datasets, training regimes and architectures play a part in the formation of reasoning gaps, it is generally assumed that all three choices are relevant (Nerem et al., 2025). Furthermore, recent evidence strongly implies that specific LLM architecture choices—such as causal masking (Barbero et al., 2024a), softmax attention (Veličković et al., 2024) and positional embeddings (Barbero et al., 2024b) can directly cause such gaps to appear.

To simulate how a model might adapt to unfamiliar situations, many studies dedicated to robustifying reasoning in LMs will evaluate out-of-distribution performance (typically *length generalisation*) on synthetic tasks (Anil et al., 2022; Zhou et al., 2022, 2023; Ruoss et al., 2023; Zhou et al., 2024; Sanford et al., 2024; Bounsi et al., 2024; Opper et al., 2025; Vitvitskyi et al., 2025). Critically, each of these works constructs a bespoke dataset, making it hard to evaluate the relative importance of various works. We notice that, only a few years ago, a very similar conundrum affected the evaluation of algorithmic execution capabilities of graph neural networks (Xu et al., 2019; Veličković et al., 2019; Tang et al., 2020). Therein, the issue was successfully addressed with the advent of the *CLRS* benchmark (Veličković

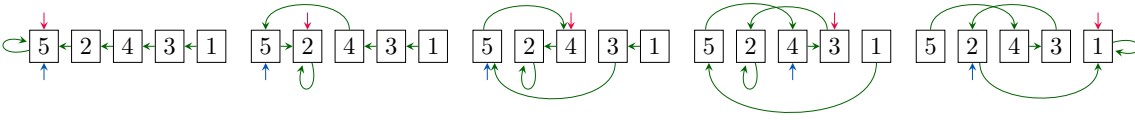

```
insertion_sort:
key:  [5 2 4 3 1], initial_trace:  [5 2 4 3 1]
trace | pred:
[2 5 4 3 1], [2 4 5 3 1], [2 3 4 5 1] | [1 2 3 4 5]
```

Figure 1: **Top:** The *graph* algorithmic trace of insertion sorting a list $[5, 2, 4, 3, 1]$ in graph form (reprinted from Veličković et al. (2022)). **Bottom:** The same algorithmic trace, represented *textually*, by using our provided CLRS-Text generator. The model receives as *input* (depicted in **green**) the input array (`key`) and the initial value of the sorting trace (`initial_trace`), using which it is prompted to predict the *trace* (depicted in **blue**) of gradually sorting the list, by inserting one element at a time into a partially sorted list, from left to right. At the end, the model needs to *output* the final sorted array (depicted in **red**), and it is evaluated on whether this array is predicted correctly.

et al., 2022)—a dataset generator comprising execution traces of thirty classical algorithms from the Introduction to Algorithms textbook (Cormen et al., 2022), while allowing for a principled way to generate traces for novel tasks, enabling several novel research directions (Ibarz et al., 2022; Bevilacqua et al., 2023; Minder et al., 2023; Jürß et al., 2024).

Our aim is to *bring the benefits of CLRS into language modelling*, yielding the **CLRS-Text** benchmark. CLRS-Text is a procedural dataset generator based on converting the graph-based traces within CLRS into textual form, making them suitable for ingestion by language models—see Figures 1–4 for specific examples in sorting (Figure 1), dynamic programming (Figure 2), path-finding (Figure 3) and string matching (Figure 4).

## 2 Motivation

To clarify why we find CLRS-Text to be an important benchmark for evaluating reasoning capabilities, we will first discuss our operational definition of reasoning, along with the implications for what a dataset for evaluating this kind of reasoning might look like.

To us, *reasoning is a **robust** procedure for solving instances of a problem*. We impose no requirement for this procedure to be fully accurate: we consider humans capable of reasoning, yet human reasoning is often approximate or relies on partial information. Further, we do not require the model to explicitly trace its computation using symbols—reasoning may be successfully done in the latent space. The key factor, *robustness*, implies the model should behave *consistently* across diverse problem instances[1]. Accordingly, we care about

---

1. Or, at least, the model should be able to give appropriate confidence estimates in its answers—or even avoid answering altogether—if the input substantially escapes the support of its training data. This is hard to reconcile with modern practices of instruction tuning (Wei et al., 2021), which effectively teaches a model to always attempt to provide answers.

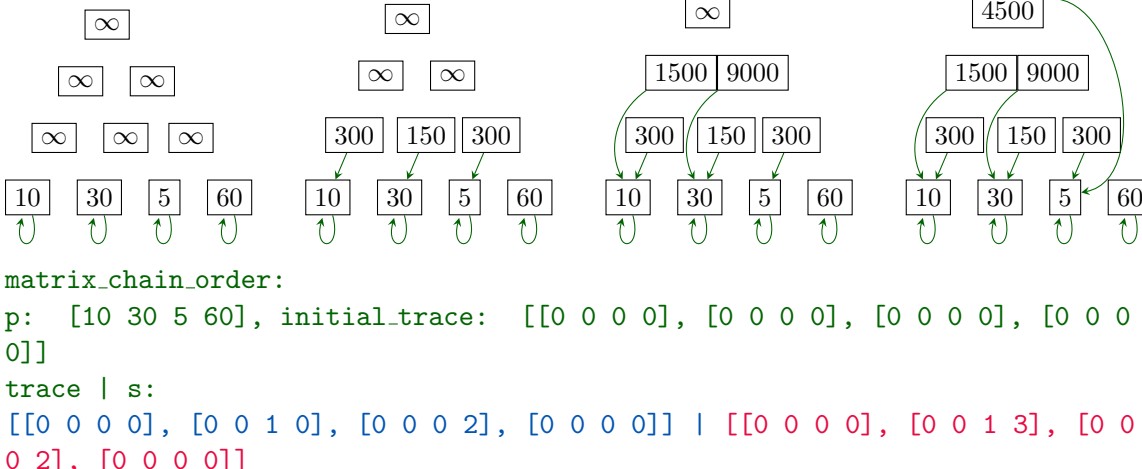

```
matrix_chain_order:
p:  [10 30 5 60], initial_trace:  [[0 0 0 0], [0 0 0 0], [0 0 0 0], [0 0 0
0]]
trace | s:
[[0 0 0 0], [0 0 1 0], [0 0 0 2], [0 0 0 0]] | [[0 0 0 0], [0 0 1 3], [0 0
0 2], [0 0 0 0]]
```

Figure 2: **Top:** The *graph* algorithmic trace of optimising the order of multiplications in a chain of matrices, for multiplying matrices of size $(10 \times 30)(30 \times 5)(5 \times 60)$, assuming a $O(n^3)$-time multiplication algorithm (reprinted from Veličković et al. (2022)). **Bottom:** The same algorithmic trace, represented *textually*, by using our provided CLRS-Text generator. The model receives the input matrix sizes (`p`) and the initial value of the pointers (`initial_trace`), using which it is prompted to predict the trace of gradually determining optimal orders of multiplying various subchains of the original chain of matrices. Note that, in our default data generator, we do not store intermediate numbers of operations—only the pointers are preserved in the trace.

*out-of-distribution* generalisation in LMs. Prior work shows many present frontier models fail to length generalise, even for relatively simple arithmetic operations such as multiplication (Dziri et al., 2023; Shen et al., 2023).

How can we evaluate out-of-distribution generalisation? Firstly, note that full OOD is hard to encounter or even construct with modern pre-training setups, wherein LMs are trained on Internet-scale data (though we will provide results of pre-trained frontier LMs on CLRS-Text as an indication of their immediate algorithmic reasoning capabilities). Because of this, we argue for hand-crafting specialised data that we will finetune models on, and also construct appropriate IID and OOD evaluation datasets for. This is standard practice in several relevant prior works (Delétang et al., 2022; Ruoss et al., 2023; Shen et al., 2023).

In order to be able to do this automatically, we need to focus our attention on tasks that allow outputs to be generated (a) reliably, (b) efficiently, and (c) for any valid input distribution of interest. These constraints, taken together with the expectation that reasoning should be a robust procedure, imply that we should train and evaluate our models on traces of **polynomial-time algorithms**, which exactly correspond to robust, well-defined procedures that perform their computations in a tractable manner.

This is exactly what the CLRS benchmark is designed to do—expose traces of polynomial-time algorithms in a rigid, efficient and unified manner, for any valid input distribution. It

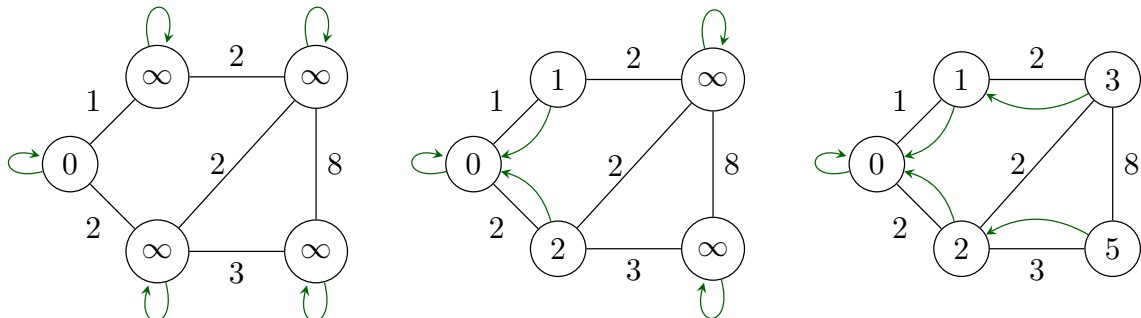

```
bellman_ford:
s:  0, A: [[0 1 2 0 0], [1 0 0 2 0], [2 0 0 2 3], [0 2 2 0 8], [0 0 3 8
0]], initial_trace:[0 1 2 3 4]
trace | pi:
[0 0 0 3 4] | [0 0 0 1 2]
```

Figure 3: **Top:** The *graph* algorithmic trace of finding single-source shortest paths (from node zero) using the Bellman-Ford algorithm, for a given undirected weighted graph (reprinted from Veličković et al. (2022)). **Bottom:** The same algorithmic trace, represented *textually*, by using our provided CLRS-Text generator. The model receives the source node identity (`s`), the weighted adjacency matrix (`A`) and the initial value of the predecessor pointers (`initial_trace`), using which it is prompted to predict the trace of gradually recomputing predecessor pointers until all single-source shortest paths are found. Note that, in our default data generator, we do not store intermediate path lengths—only the pointers are preserved in the trace.

is this observation that fuels our decision to extend CLRS into the textual domain and train language models on its traces. It is our hope that CLRS-text has the potential to become an important benchmark for reasoning, precisely because it allows for easy generation of bespoke trace data at various distributions, and simplifies setting up comparisons across multiple papers that use it.

Lastly, and conveniently: since we evaluate our model on procedurally-generated test data, we can constantly *resample* the test datapoints, ameliorating risks of reasoning gaps from hill-climbing static datasets (Srivastava et al., 2024; Mirzadeh et al., 2024).

Before presenting some motivating results, we want to make note of a recent important initiative: the GraphArena dataset (Tang et al., 2025). In many ways, GraphArena is a challenge compatible with CLRS-Text's aims, as it evaluates LLM capabilities on graph-structured tasks with a robust evaluation protocol. However, GraphArena is derived from real-world scraped graphs rather than a resampled dataset, which limits the extent to which generalisation might be tested—and explicit traces are not provided.

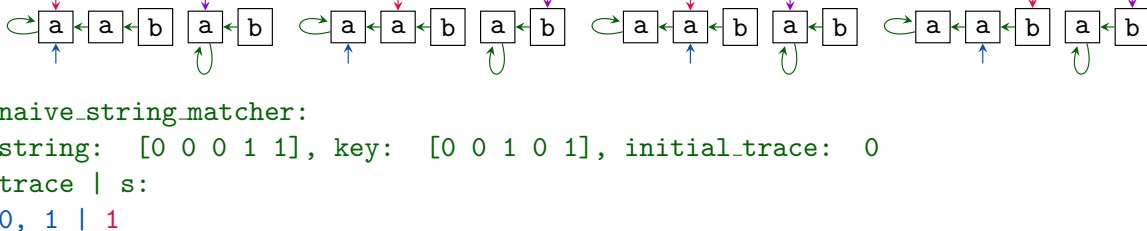

```
naive_string_matcher:
string:  [0 0 0 1 1], key:  [0 0 1 0 1], initial_trace:  0
trace | s:
0, 1 | 1
```

Figure 4: **Top:** The *graph* algorithmic trace of finding the first occurence of the string `ab` inside the string `aab` (reprinted from Veličković et al. (2022)). **Bottom:** The same algorithmic trace, represented *textually*, by using our provided CLRS-Text generator. The model receives the identifier of which character belongs to which string (`string`), the value of each character (`key`) and the initial value of the position at which the match is queried (`initial_trace`). Using this information, the model is prompted to predict the trace of gradually adjusting the querying position until a full match is found.

## 3 CLRS-Text construction

As already mentioned, CLRS-Text is entirely based on the representations computed by the CLRS benchmark (Veličković et al., 2022). For a brief recap: each algorithm in CLRS is specified by *inputs*, *traces*[2] and *outputs*, and by default, CLRS offers access to thirty classical algorithms from Cormen et al. (2022), spanning sorting, searching, divide & conquer, greedy algorithms, dynamic programming, graph algorithms, string algorithms and geometric algorithms.

For each sample of every algorithm, it is assumed that inputs and outputs are kept fixed, whereas the traces represent the trajectory of (internal) states the algorithm goes through while computing the output. Since CLRS was originally designed to train non-autoregressive models, it natively leverages a *graph* representation of this data.

In contrast, CLRS-Text converts the trace data to text. How should this conversion be done? In general, this is something that users of CLRS-Text can tweak to their needs, by modifying the default converters provided in `clrs_utils.py`.

The default conversion function we provide (with outputs illustrated by Figures 1–4) is designed with *limited context windows* in mind. We would like our traces to completely fit in context of smaller-tier models such as Gemma 2B (Gemma Team et al., 2024), to allow for simplified training at a wide range of parameter scales. This means that, especially for tasks including information on edges of graphs ($O(n^2)$ entries for problems of size $n$), we cannot afford to print all parts of the algorithm's trace, and instead we focus on printing *exactly one variable*'s trace—the variable which eventually converges to the output. In the concrete case of sorting algorithms from Figure 1, the trace we print is the state of the input array after each step of the algorithm.

---

2. In the original CLRS benchmark, the term "hints" is occasionally used instead of "traces".

The only algorithm in the thirty default algorithms of CLRS for which we do not provide a trace is the segment intersection algorithm, as it has $O(1)$ time complexity and therefore does not have atomic steps that converge to the final output.

Note that, because we do not print the entirety of the algorithm's state at every step, it is possible that the trace may remain *unchanged* in certain steps—for example, when insertion sorting an already-sorted array of $n$ elements, each of the $n$ steps of the trace will be identical. We argue that it is *useful* to encourage the model to produce such traces, as it explicitly indicates to the model the likely "thinking time" needed for solving the task through chain of thought (Wei et al., 2022). Further, recent results indicate that the mere amount of chain of thought tokens may be correlated with the relative increase in expressive power of Transformer-based language models (Merrill and Sabharwal, 2023).

Lastly, while CLRS-Text provides the same thirty algorithms present in CLRS by default, owing to the strong synergy of the two benchmarks, adding new tasks to CLRS-Text's generator requires no significant added effort compared to adding them to CLRS.

The reader interested in adding new tasks may wish to consult Appendix A of Veličković et al. (2022) for an overview of key ways to interact with CLRS—these will apply for CLRS-Text as well. The only change necessary to make in our default generator script for CLRS-Text is to indicate which part(s) of the trace should be printed—and in which format—for the newly added algorithm.

## 4 Training and evaluation

For the main part of our evaluation, we follow a rigorous out-of-distribution setup, wherein we pre-train a Gemma 2B model (Gemma Team et al., 2024) using the standard next-token prediction objective on specific problem sizes across all thirty algorithms provided with CLRS by default. In the spirit of foundation models (Bommasani et al., 2021), our pre-training follows the style of a *generalist reasoner*—building a single multi-task model capable of executing all thirty algorithms from textual prompts simultaneously.

To demonstrate how CLRS-Text can help assert established results, we also pre-train a variant of the Gemma 2B model which uses *randomised* positional embeddings (RPE)—already shown by Ruoss et al. (2023) to yield better length generalisation for tasks along the Chomsky hierarchy.

This compares to relevant multi-task learning work on graph-based CLRS (Ibarz et al., 2022; Georgiev et al., 2024; Bohde et al., 2024; Li et al., 2024), with a representational advantage in the language case: since the inputs and outputs are just textual tokens in CLRS-Text, exactly the same language model architecture can be used for all thirty tasks—whereas for the former papers, different encoder and decoder functions were necessary due to data and shape discrepancies.

Once trained, we evaluate our models zero-shot on randomly sampled CLRS-Text instances for each of the thirty algorithms, at every problem size that would fit in the model's context window, up to a maximum size of $n = 64$. For each evaluation prompt, we extract the final array-like object that the language model produced in its output, and compare it to the ground-truth prompt via exact string match.

Note that resampling allows us to directly take into account the reasoning gap across multiple runs (Srivastava et al., 2024), as we will never evaluate on static test data. This

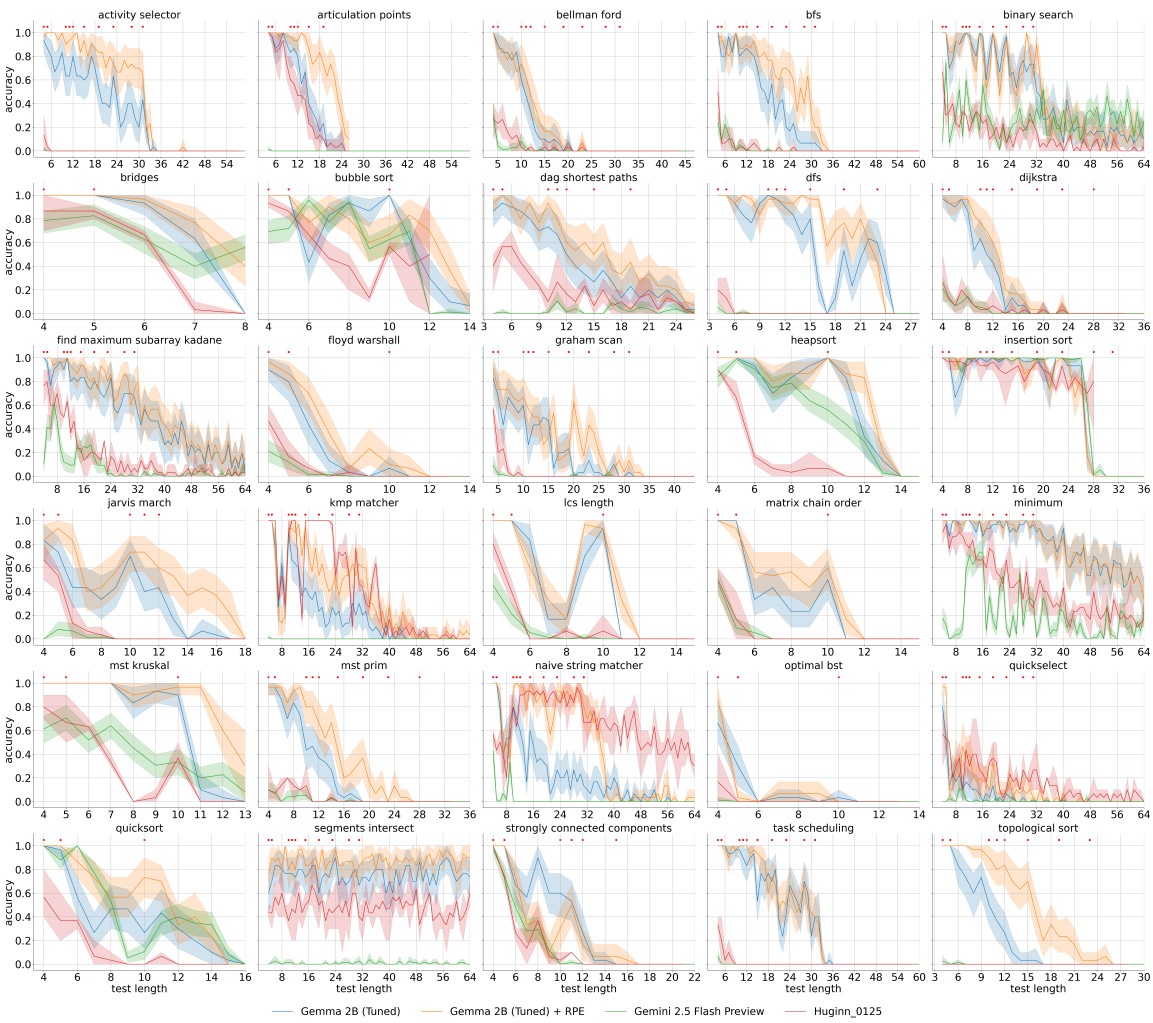

Figure 5: Resampling test results of variants of Gemma 2B, Gemini 2.5 Flash Preview and Huginn-0125, on various problem sizes. Gemma 2B variants were explicitly trained on CLRS-Text tasks—the training set sizes are denoted by red dots—and are evaluated zero-shot. Huginn-0125 used a training mixture including CLRS-Text traces, and is therefore also evaluated zero-shot. Note that Huginn-0125 has a more limited context window and therefore not all evaluation data points are provided for it. Gemini 2.5 Flash Preview is a pre-trained general-purpose model, evaluated in a two-shot manner with an instruction provided.

will evaluate both generalisation on in-distribution (training) problem sizes, interpolating out-of-distribution sizes and extrapolating out-of-distribution sizes.

For all our experiments, tool use (in the style of Schick et al. (2024)), as well as the usage of code interpreters, is explicitly *switched off*. This is due to the fact we would like to evaluate and improve *base model* robustness capabilities, without any confounding effects stemming from the tool's capabilities.

These confounding effects could be particularly pronounced in the CLRS context, as the tasks from Introduction to Algorithms are standard hallmark of a computer science undergraduate degree—hence, their implementations are omnipresent on GitHub, and it is very likely that most frontier models have been trained on such source codes. We direct interested readers to McLeish et al. (2024) for a detailed overview of the kinds of results achievable on a CLRS-Text variant when using a model with access to a code interpreter.

## 5 Results and Discussion

CLRS-Text lends itself naturally to many different forms of language model evaluation. We conduct one such evaluation, to provide an indication of how difficult it is to reliably incorporate algorithmic operations within language models.

We train two variants of the Gemma 2B model on all thirty tasks, using next-token prediction, at various training sizes. The first variant is identical to the original Gemma 2B, whereas the other employs randomised positional embeddings (Ruoss et al., 2023) with $L = 10,000$ (compared with Gemma 2B's context length of $N = 8,192$).

The training sizes (see Table 1 in the Appendix) are carefully chosen for each algorithm such that we can evaluate generalisation in both the interpolation and extrapolation regimes, taking into account Gemma 2B's context length. We would intuitively expect interpolating to be easier for the model, as it implies input context lengths that mostly have been observed during training (see Bounsi et al. (2024) for more empirical evidence). Note that only a single model is trained across all thirty tasks—mirroring the multi-task setup of Ibarz et al. (2022) for the domain of text.

The final pre-training set used by Gemma 2B in our experiments is constructed by taking $10,000$ randomly-sampled prompts of each training size.

Evaluation is performed zero-shot, recording average accuracy across three resampled test sets of 25 samples for every problem size fitting in context. This evaluation set size is deliberately chosen to trade off the robustness of the evaluation with the computational complexity of running the evaluation on a typical GPU setup. To verify this, we have also performed an evaluation on five resampled test sets of 125 samples per problem size – which would take multiple days on an easily-accessible GPU inference setup – and found that our takeaways remained consistent, albeit with reduced variance. Each example is assessed using exact string match on the final array produced.

While general-purpose frontier models are not necessarily out-of-distribution with respect to these problems or even these sizes, we believe it is useful to report the performance of such models on our test sets, to be able to internalise the gap these models have to cross before they will be capable to solve CLRS-Text tasks reliably. For this purpose, we include two-shot evaluation results (with a provided instruction) for Gemini 2.5 Flash Pre-

view (Google, 2025), which is a fast version of the frontier model Gemini 2.5 Pro Preview. We disable the Thinking mode when performing this evaluation.

In addition, we evaluate the recent Huginn-0125 foundation model (Geiping et al., 2025) which has been explicitly trained on CLRS-Text traces as part of its training data mixture. As such, this model is evaluated zero-shot. While Huginn uses a less standardised *latent reasoning* architecture leveraging recurrent layers, this comparison should indicate the extent to which CLRS-Text knowledge is retained as part of a broader pre-training phase.

We provide evaluation results across all thirty algorithms in Figure 5. Our results are indicative that the benchmark should hold utility for evaluating robust reasoning of base model architectures, several of which we enumerate below:

- Even though the computations encompassed in Cormen et al. (2022) are standard constructs in computer science – easily found all across the Internet – there is a clear lack of internalising this knowledge by the pre-trained general-purpose model, as it is not capable of achieving comparable results to the significantly weaker but fine-tuned Gemma 2B on most algorithms[3].

- While some parts of this knowledge can be preserved by including CLRS-Text within a general pre-training mixture (as done by Huginn-0125), the benefits do not persist across most algorithms. A notable exception to this are string matching algorithms (naïve string matcher and KMP matcher), which rely on understanding string data and therefore might benefit from either (a) the language biases in Huginn-0125's more general pre-training mixture, or (b) Huginn-0125's explicit use of recurrent layers.

- Our results indicate that CLRS-Text can be an excellent testbed for targeted model modifications, exposing clear differences between various approaches when relevant. Specifically, the use of randomised positional embeddings clearly improves generalisation over the baseline Gemma 2B—as shown by Ruoss et al. (2023). However, the effects of this modification still taper off quickly in the extrapolation regime, indicating that more substantial changes might be necessary to make meaningful progress on this benchmark in the OOD regime.

Extending from the final point, it is worth noting that, on the graph variant of CLRS, multi-task reasoners easily generalise to $4\times$ the input sizes seen at training time (Ibarz et al., 2022), whereas on CLRS-Text, all of the language models we attempted barely extrapolate out-of-distribution at all. We suspect that the autoregressive nature of LLMs is to blame (Barbero et al., 2024a)—for reasoning problems, one can often predict multiple parts of the output at once rather than having to predict them one token at a time, which GNNs can easily exploit. We highlight this as an important future work direction.

In all cases except sorting, the pre-trained general-purpose model is not capable of achieving comparable results to the fine-tuned Gemma 2B, indicating there exists a clear improvement in general-purpose algorithmic reasoning which is attainable with smaller models, that we may manage to make available to our frontier models in the future.

Note that, even though the out-of-distribution performance of current language models on CLRS-Text is not satisfactory, it should *not* be seen as evidence that "language models

---

3. A notable exception are sorting algorithms.

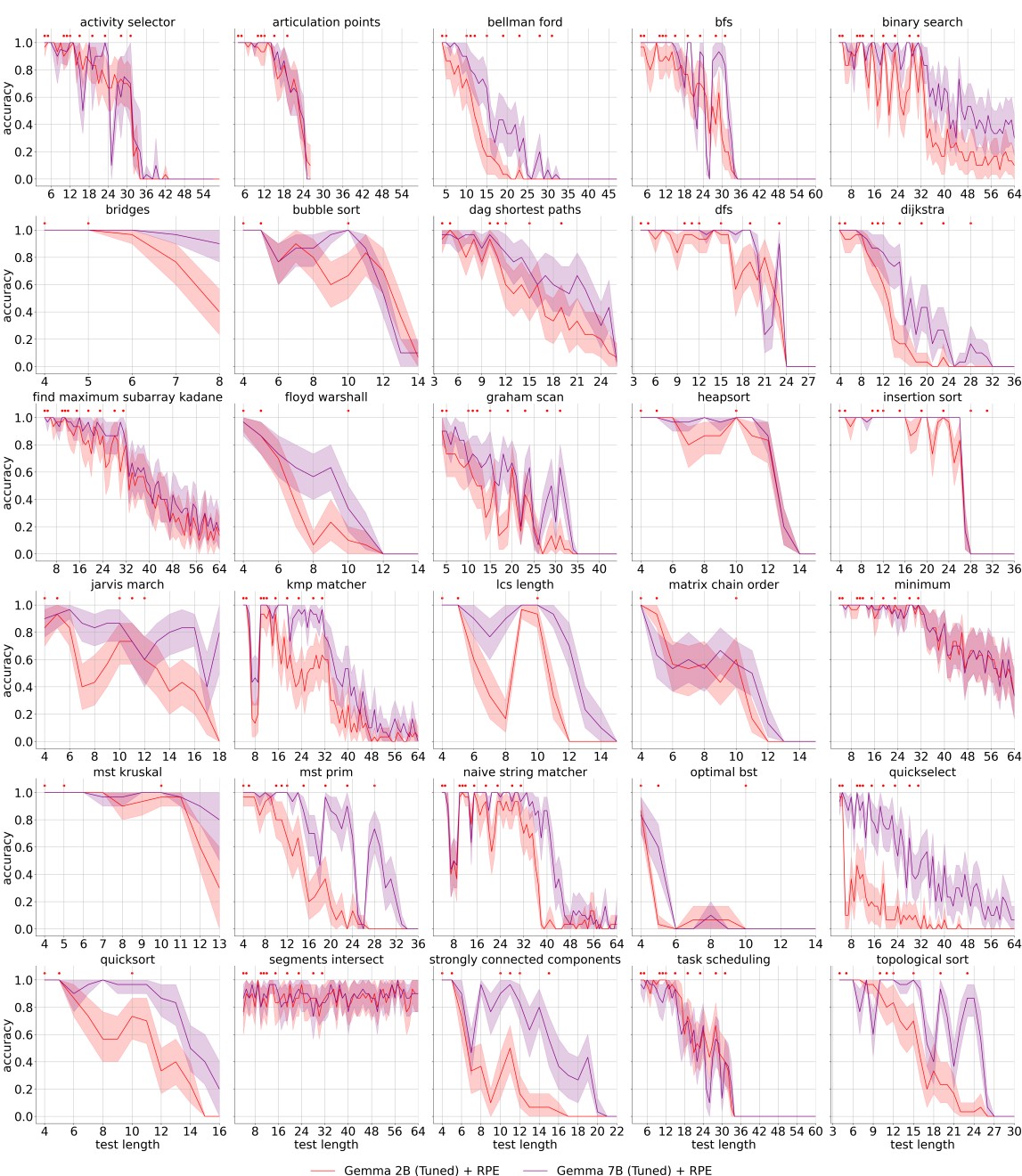

Figure 6: An evaluation comparing Gemma 2B and Gemma 7B (both leveraging RPE) revealed that a 3.5× increase in model size generally led to improved performance. This upscaling particularly benefited the fitting of results for difficult algorithms, including KMP matcher, strongly connected components, topological sorting, quickselect and quicksort. Nevertheless, as the task length deviates significantly from the largest training sizes, the quality of the results consistently degrades, irrespective of the model's overall parameter count.

cannot reason". We test for a very specific kind of reasoning here—namely, algorithmic generalisation to distribution shifts—and emphasise that not all reasoning behaviours require such capability; for example, we may fully know upfront all the feasible input distributions for a particular reasoning task, and generate appropriate data to focus only on in-distribution generalisation—side-stepping our setup entirely.

## 6 Ablation studies

Our results naturally invite many possible follow-up studies that would be aimed to better understanding the causes behind the collapsed out-of-distribution behaviours. Furthermore, our CLRS-Text platform may be readily used to set up many different kinds of data / model combinations than the ones studied here. While we cannot hope to exhaustively cover all of those settings in this paper, we will now highlight *three* specific interesting research questions for which we provide concrete ablation studies. These results will finalise our paper's contributions and hopefully invite ample further work.

**Can *scale* help?** (A: Yes, but not in a way that changes the OOD trends.)

Given the many successes of large LMs—especially with scaling up their capacity—it is a natural question to examine whether adding more *scale* to the model parameters makes a meaningful difference to the trends observed in our previous experiments.

As an exemplar for this kind of experiment, in Figure 6 we compare tuned Gemma 2B with Gemma 7B (both with RPE); this is convenient because the base architectural choices are fixed, and we modify only the scale at which the architecture is generated.

Our results indicate that increasing scale clearly improves the model's evaluation performance across the range of algorithms and sizes. In particular, adding capacity clearly enables the model to fit certain difficult algorithms in-distribution a lot better, as one would arguably expect.

That being said, we find that scale generally does not change our overall takeaways—as we go even modestly beyond the largest training size, the performance still tends to collapse towards zero. This indicates that the generalisation performance issues in CLRS-Text are likely to be significantly caused by architectural decisions, which, in turn, should make this a useful benchmark for architectural innovation.

**Are the algorithms *actually followed* at all?** (A: Yes, in most cases.)

Another issue that may very reasonably be raised with our evaluation setup is that, as we only measure the *final step accuracy*, we are not truly measuring to what extent the model learnt the target algorithm (as expressed in the printed trace), as opposed to simply producing its outputs in a different manner.

In order to meaningfully assess the extent to which the algorithms are learned, in Figure 7 we plot, for the tuned Gemma 2B model with RPE, the "time-to-first-failure" metric, which shows how much of the intermediate trace (expressed as a fractional value) is fitted correctly by the LM before a mistake is encountered. Higher values of this metric imply a larger degree of alignment between the model's produced trace and the target algorithm's,

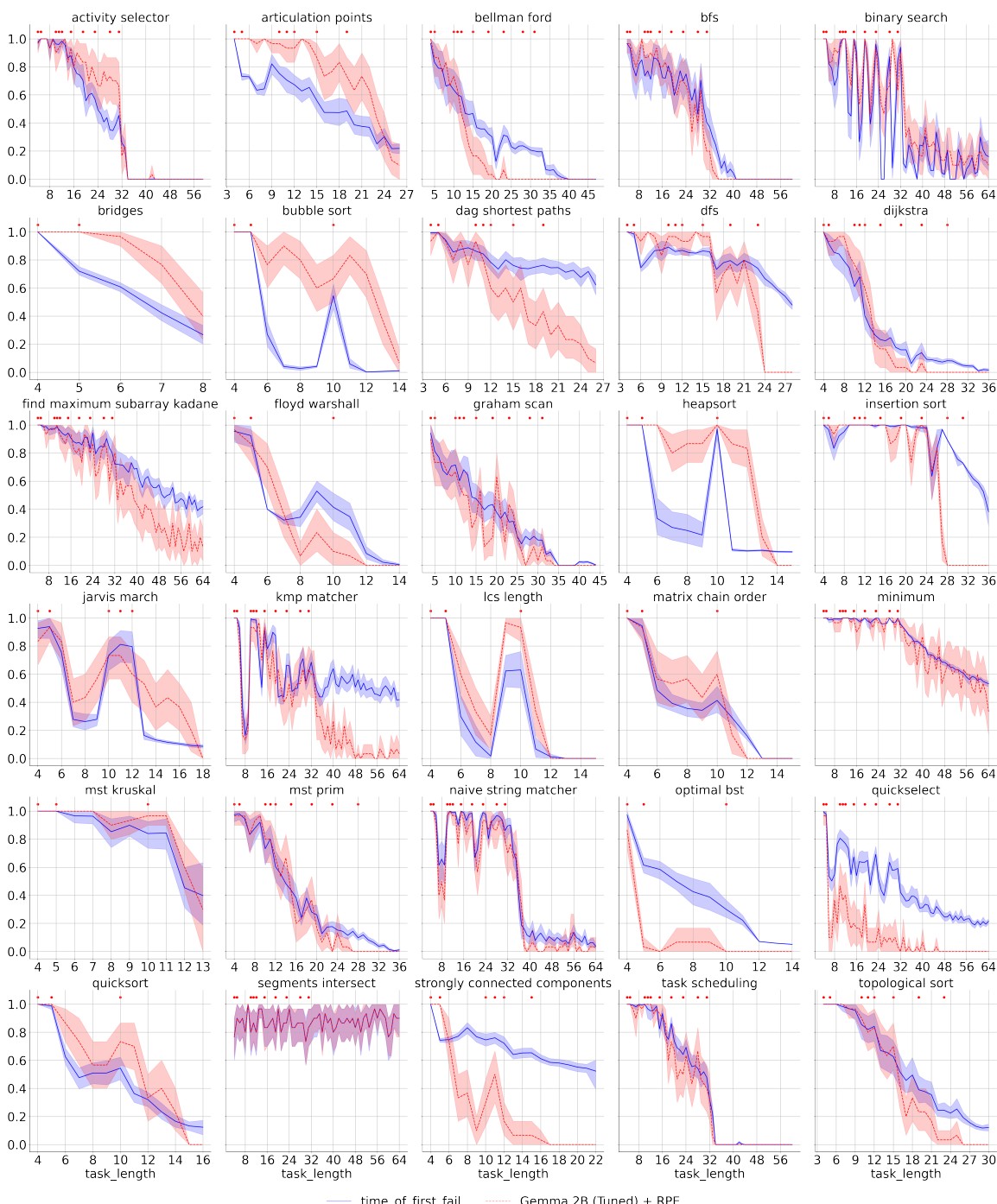

Figure 7: This image compares our accuracy metric with *time-to-first-failure*, i.e., the average (fractional) step where the model's answer starts diverging from the ground truth trace. Generally, time-to-first-failure surpasses and follows the final accuracy's trend. However, exceptions like bridges, bubble sort and heapsort exist. This suggests the model typically learns to follow the algorithms it trained on step-by-step, but for algorithms such as bubble sort, it appears to 'jump straight to the conclusion' rather than performing its operations sequentially.

generally indicating that the algorithm has been learnt better on a given input[4]. We then plot this metric against input size, side-by-side with the final accuracy plots from before. This allows us to correlate the model's algorithmic instruction against its accuracy.

Overall, our results imply that in most cases, the performance profile of the time-to-first-fail metric meaningfully corresponds to the accuracy metric's, and furthermore it is usually equal or higher than the accuracy. Both of these taken together imply that the model has indeed learnt a good degree of algorithmic instruction following, and that this is a good predictor of final performance.

Two notable exceptions to this include *challenging detections of graph substructures* (namely, articulation points and bridges) and *particular sorting algorithms* (bubble sort and heapsort). In both of these cases, the model retains reasonably high performance while its time-to-first-fail metric is erratic and suboptimal. This implies that the model might have simply found simpler heuristics for the detecting graph substructures, and/or that it has simply learned to sort in a different way than the ones suggested by bubble sort and heapsort.

**What about *other distribution shifts*?** (A: The ones we tried are even *worse*.)

Our work, like many other out-of-distribution generalisation works preceding it, focusses on length generalisation, as it is both among the simplest distribution shifts to reason about *and* applicable to arbitrary algorithmic tasks. However, it is far from the only possible distribution shift, and one might naturally wonder how well do tuned language models perform when tasked with other generic distribution shifts.

To exemplify such shifts, we evaluate our tuned Gemma 2B model with RPE on two additional test datasets (without retraining), featuring additional distribution changes:

- Modifying the *graph* distribution used for graph tasks, from Erdős-Rényi (ER) graphs to random *trees*. ER graph distributions are generally well-connected and low diameter for many choices of their parameters; in contrast, trees have ample bottlenecks and higher diameter, necessitating long execution traces (Georgiev et al., 2023).

- Modifying the *value* distribution for all scalar values, from $U(0,1)$ to $U(1,2)$. Value shifts typically expose shortcomings in the arithmetic learned by algorithm executors, as exemplified by, e.g., Mirjanić et al. (2023); de Luca et al. (2024).

The results of this study are summarised in Figure 8, and they reveal that these shifts are indeed sufficient to further degrade performance of the tuned LM executor—sometimes catastrophically so. Moreover, the specific degradations we observe nicely correspond to the expectations of whether particular algorithms leverage data susceptible to these shifts:

- String matching algorithms (naïve string matcher and KMP matcher) and the longest common subsequence (LCS) algorithm are unaffected by either shift—as they do not rely on scalar values *or* graphs.

---

4. Note that we include the final output within the trace for the purposes of this analysis, therefore a perfect score on time-to-first-failure implies the example has been correctly solved. The converse is not true (it is possible to have arbitrary time-to-first-fail and still solve the problem correctly).

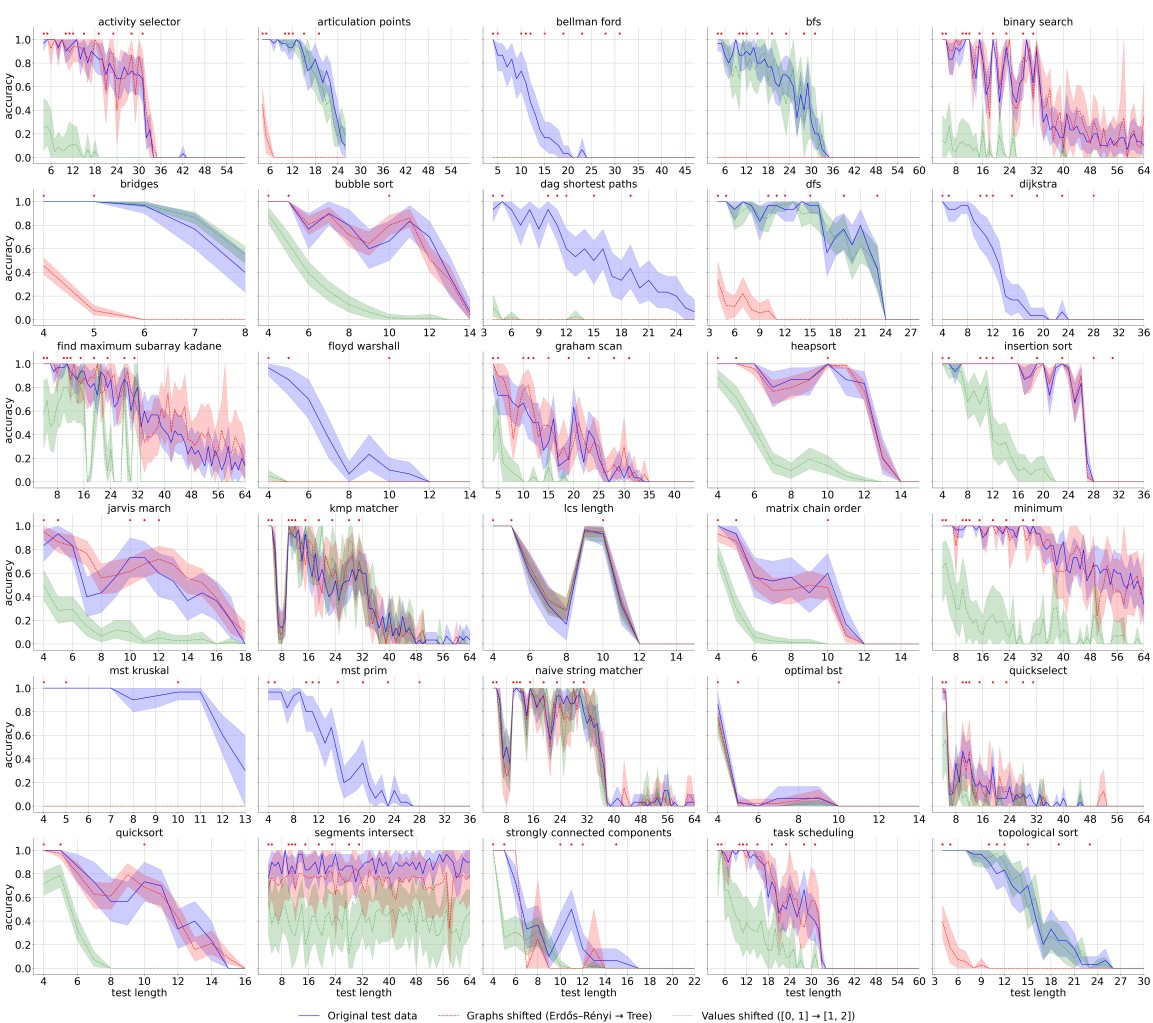

Figure 8: The figure presents the Gemma 2B (with RPE) model's performance under various evaluation data distribution shifts: the original dataset (as in Figures 5–7), replacing the Erdős-Rényi graph distribution with random trees, and shifting the range of scalar inputs. Critically, any alteration to the original data distribution consistently degraded the quality of the model's answers.

- Algorithms involving sorting and searching in arrays (activity selector, binary search, bubble sort, find maximum subarray, heapsort, insertion sort, minimum, quickselect, quicksort and task scheduling), dynamic programming structures (matrix chain order and optimal BST), as well as geometric algorithms over point clouds (Graham scan, Jarvis' march and segments intersect) are affected by the value shift (affecting array elements or point coordinates), but not by the graph shift (as they do not rely on explicit graphs).

- Conversely, unweighted graph algorithms (articulation points, BFS, bridges, DFS, strongly connected components and topological sorting) are affected by the graph shift but not by the value shift.

- Finally, algorithms operating over weighted graphs (Bellman-Ford, DAG shortest paths, Dijsktra, Floyd-Warshall, Kruskal's and Prim's MST algorithms) are affected by *both* distribution shifts – and often drastically so, as they require coherence over longer execution traces with fewer opportunities for shortcutting.

These results neatly illustrate that issues of out-of-distribution performance recur way beyond simple length generalisation, and also the role benchmarks like CLRS-Text can play in future research towards ameliorating them.

## Broader Impact Statement

CLRS-Text benchmarks Language Models (LMs) by evaluating their ability to execute and generalize from textual representations of classical algorithms. This work could contribute to more robust LMs, especially in applications needing complex reasoning. The standardized nature of CLRS-Text allows for consistent evaluation and comparison of different LMs, potentially accelerating progress. Further, our benchmark lowers the barrier of entry for research that systematically improves algorithmic reasoning in LMs, potentially enabling any dual-use applications that require such generalisation.

## Acknowledgments and Disclosure of Funding

CLRS-Text was developed with kind support of Ethan Dyer and Behnam Neyshabur, whom we are very grateful to. We are also indebted to Alek Andreev, for his kind support in setting up training for the models in the Gemma family. Lastly, we thank Razvan Pascanu, Simon Osindero and Slav Petrov for reviewing the paper prior to submission.

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

| Algorithm | Training sizes |
| --- | --- |
| Articulation points | $[4, 5, 10, 11, 12, 15, 19]$ |
| Activity selector | $[4, 5, 10, 11, 12, 15, 19, 23, 28, 31]$ |
| Bellman-Ford | $[4, 5, 10, 11, 12, 15, 19, 23, 28, 31]$ |
| Binary search | $[4, 5, 10, 11, 12, 15, 19, 23, 28, 31]$ |
| Breadth-first search | $[4, 5, 10, 11, 12, 15, 19, 23, 28, 31]$ |
| Bridges | $[4, 5]$ |
| Bubble sort | $[4, 5, 10]$ |
| DAG shortest paths | $[4, 5, 10, 11, 12, 15, 19]$ |
| Depth-first search | $[4, 5, 10, 11, 12, 15, 19, 23]$ |
| Dijkstra | $[4, 5, 10, 11, 12, 15, 19, 23, 28]$ |
| Find Maximum Subarray | $[4, 5, 10, 11, 12, 15, 19, 23, 28, 31]$ |
| Floyd-Warshall | $[4, 5, 10]$ |
| Graham scan | $[4, 5, 10, 11, 12, 15, 19, 23, 28, 31]$ |
| Heapsort | $[4, 5, 10]$ |
| Insertion sort | $[4, 5, 10, 11, 12, 15, 19, 23, 28, 31]$ |
| Jarvis' march | $[4, 5, 10, 11, 12]$ |
| Kruskal's algorithm | $[4, 5, 10]$ |
| Knuth-Morris-Pratt | $[4, 5, 10, 11, 12, 15, 19, 23, 28, 31]$ |
| Longest common subsequence | $[4, 5, 10]$ |
| Matrix chain multiplication | $[4, 5, 10]$ |
| Minimum | $[4, 5, 10, 11, 12, 15, 19, 23, 28, 31]$ |
| Naïve string matcher | $[4, 5, 10, 11, 12, 15, 19, 23, 28, 31]$ |
| Optimal binary search tree | $[4, 5, 10]$ |
| Prim's algorithm | $[4, 5, 10, 11, 12, 15, 19, 23, 28]$ |
| Quickselect | $[4, 5, 10, 11, 12, 15, 19, 23, 28, 31]$ |
| Quicksort | $[4, 5, 10]$ |
| Segments intersect | $[4, 5, 10, 11, 12, 15, 19, 23, 28, 31]$ |
| Strongly connected components | $[4, 5, 10, 11, 12, 15]$ |
| Task scheduling | $[4, 5, 10, 11, 12, 15, 19, 23, 28, 31]$ |
| Topological sort | $[4, 5, 10, 11, 12, 15, 19, 23]$ |

Table 1: The training sizes employed for generating the pre-training set used for Gemma 2B models in our study. Note that, for Segments intersect, the size parameter does not influence the generated prompt.

