# OpenReview forum: "The CLRS-Text Algorithmic Reasoning Language Benchmark"
_DMLR — Accepted by DMLR_

### Review · Reviewer_DYWu · 2025-08-12

**Recommendation:** 3
**Confidence:** 2

**Summary Of Contributions:**

This paper introduces CLRS-Text, a procedural dataset generator that converts the graph-based execution traces from the CLRS benchmark into textual form, making them suitable for training and evaluating large language models on algorithmic reasoning tasks. The benchmark covers 30 classical algorithms from Introduction to Algorithms, supports arbitrary input distributions and problem sizes, and enables out-of-distribution (OOD) evaluation via interpolation and extrapolation in task sizes. The authors train Gemma 2 (with and without randomized positional embeddings) and compare with Huginn-0125 and Gemini 2.5 Flash, evaluating zero-shot performance on resampled test sets.

**Strengths:**

See above.

**Audience:**

Yes

**Broader Impact Concerns:**

None.

**Claims And Evidence:**

Yes.

**Datasets And Benchmarks:**

Yes.

**Extended Submissions:**

Yes.

**Limitations:**

See above.

**Requested Changes:**

The detailed discussion of the results should be included.

**Strengths And Weaknesses:**

**Strengths:**

*1. Dataset generation approach reduces leaderboard saturation.* By procedurally generating data and continuously resampling test sets, the benchmark avoids the overfitting and saturation issues commonly seen with static datasets.

*2. Extension of CLRS to LLMs.* The work successfully adapts the widely-used CLRS algorithmic reasoning benchmark from GNNs to textual form, enabling direct application to language models without the need for specialized graph encoders or decoders.

**Weaknesses:**

*1. Limited exploration of cross-task generalization.* While interpolation and extrapolation in input size are a form of OOD evaluation, the paper does not explore cross-task generalization, which is considered a more general OOD setting. For example, can a model trained on Kruskal generalize to Prim, or from Bellman-Ford to Dijkstra’s shortest path algorithm? This would be highly relevant for assessing transfer between algorithmic variants solving the same problem.

*2. Model scale limitation.* The experiments focus on Gemma 2B. Given that reasoning ability often follows scaling laws, it would be valuable to explore how larger models perform on CLRS-Text, especially in the OOD regimes.

*3. Unexplained result anomalies.*  From the reported results, Huginn-0125 underperforms tuned Gemma 2B on most tasks, as expected from the setting. However, it achieves the best performance in Quickselect, Naive String Matcher, and competitive performance on KMP. Authors should discuss why these exceptions occur.

*4. Limited algorithm diversity.* Although the framework can generate datasets from any algorithm function and associated metadata, the default set is still limited to the original CLRS algorithms, most of which are graph algorithms. It would strengthen the benchmark to include more diverse algorithm classes, such as dynamic programming tasks (beyond shortest path algorithms).

**Suggestions:** Given that few-shot learning has been shown to substantially improve reasoning performance in LLMs, it would be worthwhile to explore few-shot settings for CLRS-Text tasks in addition to the zero-shot evaluations presented.

---

### Review · Reviewer_XvJw · 2025-08-17

**Recommendation:** 4
**Confidence:** 2

**Summary Of Contributions:**

- Extends CLRS-30 (graph-based) benchmark into the textual domain, making it suitable to evaluate reasoning capabilities of language models (LMs)
- The proposed extension allows unlimited resampling of prompts at arbitrary input distributions and problem sizes, unlike static reasoning benchmarks (e.g. GSM-8K, MATH)
- Empirical study showing the generalization gaps present in LMs on algorithmic reasoning tasks. Specifically, the authors show that small models like Gemma-2B, when fine-tuned on CLRS-Text, can successfully execute all 30 algorithms in-distribution but struggle to generalize to unseen input sizes, while frontier LMs (Gemini 2.5, Huginn-0125) perform worse despite massive scale, revealing a weakness of current language models in algorithmic reasoning

**Note:** Score increased after rebuttal by one point. (3->4)

**Strengths:**

See Strengths and Weaknesses.

**Audience:**

Yes

**Broader Impact Concerns:**

- Poor performance on CLRS-Text could be overgeneralized as evidence that LMs "cannot reason," even though the benchmark specifically targets algorithmic generalization
- More robust algorithmic reasoning in LMs could have both positive and dual-use risks. The benchmark lowers barriers to systematically improving algorithmic reasoning in LMs, so explicit mention of these dual-use possibilities would strengthen the impact section

**Claims And Evidence:**

For the most part, yes. A caveat is them positing the autoregressive limitation as a likely cause for poor extrapolation, when their experimental regime does not support this directly (see the above comments on how finetuning is only done on 2B, and the suggestion to also finetune on 7B).

**Datasets And Benchmarks:**

CLRS-Text is not a scraped corpus but a procedural generator, derived from the already-established CLRS benchmark (graph traces of 30 algorithms). CLRS-Text provides sufficient detail for data collection, organization, availability, maintenance, and reproducibility.

**Extended Submissions:**

This work was a previous workshop submission, and meets the eligibility criteria.

**Limitations:**

See Strengths and Weaknesses.

**Requested Changes:**

- State more precisely if the *reasoning gap* has to do with pretraining of frontier LMs falling short, or if it's a stronger architectural hypothesis. The experiments done suggest the former rather than the latter, but yet in their Results and Discussion section the authors mention they suspect the autoregressive architecture might be a limiter.
- In addition to final exact match, authors could consider reporting trace-step accuracy to see whether failures happen early or only at the end.

**Strengths And Weaknesses:**

**Strengths**:
- CLRS-Text makes a well-established reasoning benchmark (CLRS-30) directly usable with LMs, filling an important gap in the literature by way of a procedurally-generated reasoning benchmark
- The in-distribution vs OOD (interpolation, extrapolation) evaluation protocol is carefully designed
- The study reveals that small fine-tuned models (Gemma-2B) can solve CLRS-Text in-distribution but fail to generalise OOD, while frontier LMs perform worse overall, highlighting a real reasoning gap in current LMs

**Weaknesses**:
- The main contribution is essentially a serialization of CLRS into text; while impactful for LMs, the conceptual leap is modest
- The controlled study is done only on Gemma-2B, a relatively small model; it is unclear if the same generalization failures would persist (or lessen) with larger models fine-tuned on CLRS-Text. Additional experiments with e.g. finetuning Gemma-7B, would shift the claim from "pretraining misses algorithmic reasoning" to "LMs as an architecture can’t generalize algorithms, even when trained explicitly." Granted, given this is primarily a benchmark paper, this could be outside the present work's scope.
- The paper draws a sharp line between interpolation vs extrapolation OOD, but it is not obvious whether this distinction generalizes meaningfully to broader reasoning tasks outside this benchmark

---

### Review · Reviewer_P9Cn · 2025-08-31

**Recommendation:** 4
**Confidence:** 2

**Summary Of Contributions:**

This submission introduces CLRS-Text, a benchmark that adapts the classical CLRS algorithmic reasoning suite into a textual format for language models. The authors design a procedural data generator covering 30 algorithms across sorting, dynamic programming, shortest paths, string matching, and geometry, producing compact execution traces that fit within limited LM context windows while preserving essential algorithmic signal. Extensive experiments are conducted on the benchmarks by fine-tuning Gemma-2B variants and comparing them against state-of-the-art pretrained models (e.g., Gemini 2.5 Flash under few-shot prompting) and models trained with CLRS-Text (e.g., Huginn-0125), which highlight the necessary of the proposed benchmark.

**Strengths:**

1. This paper is clear and well written.
2. The benchmark design is novel. CLRS-Text introduces a standardized and extensible textual benchmark for algorithmic reasoning, bridging the gap between the CLRS suite and LMs.
3. Comprehensive coverage and multi-task learning. This paper covers 30 classical algorithms across diverse domains and enables a single LM to handle all tasks in a unified framework.
4. Revealing insights. This paper demonstrates that small fine-tuned models can outperform larger frontier LMs, highlights the benefit of randomized positional embeddings, and exposes autoregressive limitations in algorithmic reasoning.
5. Code and data are publicly available.

**Audience:**

Yes

**Claims And Evidence:**

Yes, the claims made in the paper are well supported by the presented evidence, though some aspects could be strengthened.

**Datasets And Benchmarks:**

Yes, in Section 3 of this paper, the authors have claimed the details of the dataset construction.

**Extended Submissions:**

N/A

**Limitations:**

See the above weaknesses and requested changes.

**Requested Changes:**

1. Consider enriching the evaluations. It would be better if this paper can involve other state-of-the-art small language models (e.g.,
Qwen2.5-1.5B)
2. Consider to add the discussions of other perspectives of OOD generalization.

**Strengths And Weaknesses:**

Strengths:
1. This paper is clear and well written.
2. The benchmark design is novel. CLRS-Text introduces a standardized and extensible textual benchmark for algorithmic reasoning, bridging the gap between the CLRS suite and LMs.
3. Comprehensive coverage and multi-task learning. This paper covers 30 classical algorithms across diverse domains and enables a single LM to handle all tasks in a unified framework.
4. Revealing insights. This paper demonstrates that small fine-tuned models can outperform larger frontier LMs, highlights the benefit of randomized positional embeddings, and exposes autoregressive limitations in algorithmic reasoning.
5. Code and data are publicly available.

Weaknesses:
1. In the experiments, only one kind of small-scale LLM, Gemma-2B, is considered, which may weaken the generalizability of the conclusions.
2. For the OOD distribution generalization, this paper mainly focuses on length extrapolation, but missing other shifts (e.g., distributional variations, input noise, symbol remapping).